# Retinal Neurochemistry

**DOI:** 10.3390/brainsci15070727

**Published:** 2025-07-08

**Authors:** Dominic Man-Kit Lam, George Ayoub

**Affiliations:** 1World Eye Organization, 1209 Shui on Centre, 6 Harbour Road, Hong Kong; 2Psychology Department, Santa Barbara City College, Santa Barbara, CA 93109, USA; neuro@sbcc.edu

**Keywords:** retina, neurochemistry, GABA, glutamate

## Abstract

The vertebrate retina is a complex neural tissue composed of a repeating array of distinct cell types that communicate through specialized synaptic connections. The neurochemistry underlying these connections reveals the synaptic chemistry, including the neurotransmitters involved and their corresponding receptors. The basic pattern of communication is that the pathway from photoreceptors to bipolar cells to ganglion cells typically uses glutamate as the signaling transmitter, with three ionotropic and one metabotropic receptor types. In contrast, much of the lateral feedback, performed by horizontal cells and amacrine cells, uses the inhibitory neurotransmitter GABA, while other amacrine cells use glycine or dopamine. This review examines all of these neurotransmitter systems for each retinal cell type, along with how these systems process the visual signals transmitted to the lateral geniculate nucleus and the visual cortex.

## 1. Introduction

While the first type of synapse observed was the neuromuscular junction, work on the vertebrate retina revealed the variety of neuro-neuronal synapses, providing a much richer understanding of synaptic communication. Indeed, as retinal research has demonstrated, the synaptic architecture varies considerably across neural systems, particularly in the vertebrate retina. The retina exhibits an array of synaptic topologies, molecular mechanisms, and functional architectures. In fact, many of these differ from the conventional presynaptic bouton-to-postsynaptic target paradigm observed in areas of the central nervous system, such as the hippocampus. Notably, the synaptic chains in the retina display unique organizational principles distinct from other neural networks. At the initial stage of retinal signaling, photoreceptors form direct synaptic connections with bipolar cells (BCs) [1,2,3,4].

Throughout this review of retinal neurochemistry, we draw from the wealth of work on vertebrates, with the goal of seeking to develop a clearer understanding of the mammalian retina from the lessons learned first in other animals. Figure 1 is provided to orient the reader as to the layout of the vertebrate retina and the various retinal layers discussed in this review, while Figure 2 provides a schematic of the cells and their connectivity.

## 2. Photoreceptors (Rods and Cones)

Rods and cones are ciliary-origin photoreceptors that use a cyclic nucleotide (cGMP) pathway for phototransduction. Rod photoreceptors are highly sensitive to low light and mediate scotopic (night) vision, while cones function in brighter light, enabling color vision and high acuity [5,6]. Both rods and cones release glutamate in darkness, decreasing release upon light activation due to hyperpolarization, which is a core aspect of their neurochemical signaling [4,5].

In darkness, photoreceptors are depolarized and continuously release glutamate. Upon light stimulation, they hyperpolarize, leading to a decrease in glutamate release. This modulation of glutamate release conveys visual information to downstream neurons.

Photoreceptor synapses primarily utilize glutamatergic transmission [7]. Glutamate release from photoreceptors occurs through continuous vesicle fusion at synaptic ribbons adjacent to the presynaptic membrane, a mechanism that enables sustained neurotransmission in response to steady depolarization. The synaptic ribbon structure significantly accelerates vesicle replenishment and release [8,9,10,11,12].

Rod and cone photoreceptors exhibit considerable variation in the number of ribbons and postsynaptic targets within their terminals. In the rods from mammalians and teleost fish, presynaptic terminals contain a small number of synaptic ribbons and engage with a limited set of postsynaptic targets, including rod BCs and horizontal cells (HCs). In contrast, cone pedicles, particularly in primates, feature multiple ribbons and establish synapses with at least ten distinct BC types and two HC types, forming a multiribbon, multitarget architecture [13,14].

Kolb [4] provides a foundational understanding of the neurochemistry within the retina, emphasizing that both rods and cones utilize glutamate as their primary neurotransmitter. This excitatory amino acid facilitates communication with bipolar cells and other downstream neurons in the visual pathway, highlighting the importance of neurotransmission in visual processing.

The development and maintenance of photoreceptors are governed by intricate molecular and transcriptional mechanisms. Critical transcription factors such as OTX2, CRX, NRL, NR2E3, RORβ, and TRβ2 are essential in specifying the fate and function of rods and cones [15]. These factors orchestrate the differentiation of retinal progenitor cells, which outlines the gene-regulatory networks that determine photoreceptor identity [16].

A comprehensive overview of the structural and functional aspects of photoreceptors shows that rods, responsible for low-light vision, and cones, which mediate color vision, exhibit distinct morphologies and phototransduction mechanisms [5]. The mentioned review also addresses the visual cycle, detailing the regeneration of the retinal chromophore and its significance for photoreceptor functionality.

The patterning and biochemical development of photoreceptors in the human retina includes the synthesis and regeneration of visual pigments (opsins) and the phototransduction cascade that allows photoreceptors to respond to light stimuli [17]. Understanding these processes is crucial in comprehending retinal health and the implications for various retinal disorders.

There are metabolic differences between rods and cones, and their unique structures necessitate distinct metabolic pathways [6]. Rods, which operate under low-light conditions, have a different energy demand compared to cones, which function in bright light and enable color discrimination. This metabolic reprogramming is essential in sustaining photoreceptor health and function, particularly in adapting to varying light environments.

The study of photoreceptors integrates aspects of neurochemistry, development, structure–function relationships, and metabolism. Continued research in these areas is essential in elucidating the complexities of vision and in developing therapeutic approaches to retinal diseases. The interactions between genetic, biochemical, and environmental factors shape the functionality and health of photoreceptors, underscoring their significance in visual perception.

## 3. Bipolar Cells

Bipolar cells exhibit at least 13 distinct types, each systematically transforming photoreceptor inputs to encode features such as polarity, contrast, temporal profiles, and the chromatic composition [18,19,20]. The neurochemical diversity among bipolar cells includes differences in neurotransmitter receptors, secondary messengers, and synaptic arrangements, affecting their signal processing and potential vulnerability to disease [20,21]. ON and OFF bipolar cells utilize different transduction cascades, with ON bipolar cells relying on a G-protein signaling mechanism to respond to light [21]. The bipolar cell output is modulated by both chemical and electrical synapses with amacrine cells, contributing to the complexity of retinal signal processing [19,22].

Bipolar cell (BC) synaptic terminals, in contrast to photoreceptors, typically form multiple ribbon synapses per terminal, targeting amacrine cells (ACs) in dyadic, triadic, or tetradic configurations. Unlike photoreceptors, BCs do not create invaginations, resulting in more spatially diffuse glutamate release. This synaptic organization leads to significant glutamate spillover, potentially facilitating NMDA receptor activation in target neurons. The overall architecture of BC ribbon synapses can thus be described as a multiribbon, semiprecise target system [14,18].

Bipolar cells serve as intermediaries between photoreceptors and ganglion cells [4,20]. They express different types of glutamate receptors:ON bipolar cells: Possess metabotropic glutamate receptors (mGluR6) that, when activated by glutamate, lead to hyperpolarization. A decrease in glutamate (due to light) results in depolarization, transmitting the signal onward.OFF bipolar cells: Contain ionotropic glutamate receptors (AMPA and kainate receptors). Glutamate binding causes depolarization; thus, reduced glutamate release during light exposure leads to hyperpolarization.

Retinal bipolar cells serve as pivotal interneurons, forming the sole direct synaptic link between photoreceptors in the outer retina and ganglion cells in the inner retina. Euler et al. [18] provide a comprehensive overview of these cells, emphasizing their functional diversity across dimensions such as chromatic tuning, response polarity (ON vs. OFF), and temporal kinetics. The authors highlight how bipolar cell function is shaped by distinct receptor expression profiles, second messenger systems, and complex synaptic mechanisms, including modulatory input from horizontal and amacrine cells that fine-tune signal integration and transformation within the retinal circuitry [18].

The biochemical and molecular diversity of bipolar cells is further elaborated upon in the work of Sherry [20], who delineates the neurochemical heterogeneity underlying bipolar cell subtypes. The analysis underscores the importance of differential neurotransmitter receptor expression and intracellular signaling pathways, which are crucial for both synaptic transmission and cellular homeostasis. Such heterogeneity is thought to contribute to the differential vulnerability of bipolar cells in retinal degenerative diseases [20].

Martemyanov and Sampath [21] delve into the transduction cascade specific to ON-type bipolar cells, focusing on the mGluR6-mediated G-protein signaling pathway in dendrites. This unique mechanism allows ON bipolar cells to depolarize in response to light, a response opposite to that of photoreceptors. The review also explores the pathological implications of disruptions in this cascade, linking molecular dysfunction to visual disorders [21].

Tsukamoto and Omi [19] contribute a detailed morphological and synaptic classification of mouse retinal bipolar cells. Special attention is given to the rod pathway, particularly the connections between rod bipolar cells, AII amacrine cells, and cone bipolar cells. Their findings underscore the structural and functional specificity of each bipolar cell type, suggesting that the rod-driven circuitry is more heterogeneous and integrative than previously appreciated [19].

A foundational understanding of bipolar cell morphology and function was provided by Kaneko [23], who established early evidence of their central role in mediating visual information between the photoreceptor and ganglion cell layers. This work laid the groundwork for later investigations into the complex synaptic and physiological properties of these interneurons [23].

Strettoi et al. [24] address the complexity of cone bipolar cells, emphasizing their functional and anatomical diversity within the cone-driven pathways. The review suggests that this heterogeneity supports the parallel processing of visual features such as contrast, color, and motion, thereby expanding the computational capabilities of the retina [24].

## 4. Bipolar Cell Receptors

The primary excitatory neurotransmitter in the retina is glutamate, which operates through ionotropic and metabotropic glutamate receptors (iGluRs and mGluRs). iGluRs include AMPA, kainate, and NMDA receptors, which function as cation-permeable channels and mediate sign-conserving excitatory transmission. AMPA receptors have two varieties, calcium-permeable AMPA receptors and calcium-impermeable AMPA receptors, with the former functioning in ON bipolar cells to provide continuous, linear contrast signaling in the rod-driven ON pathway under low and moderate light [25]. NMDA receptors, in particular, serve as coincidence detectors due to their voltage-dependent magnesium block and requirement for a co-agonist, likely D-serine from Müller glia.

In contrast, mGluR6 is uniquely expressed in ON BCs and mediates sign-inverting transmission through a G-protein-coupled cascade that closes cation channels, hyperpolarizing the cell in response to glutamate release. The divergence in glutamate receptor expression among retinal cell types contributes to the complex signal processing capabilities of the retina [4,12,26,27].

Bipolar cells universally employ glutamate as their principal excitatory neurotransmitter, serving as the main conduit for signal transmission to downstream retinal neurons, including ganglion and amacrine cells. This glutamatergic output is a defining feature across all bipolar cell subtypes. However, additional neurochemical complexity is evident. Subsets of bipolar cells contain other amino acids, such as aspartate and, in some cases, GABA or glycine, suggesting roles beyond classical excitatory signaling and pointing toward functional heterogeneity within the bipolar cell population [28]. The presence of these neurotransmitters indicates diverse mechanisms of intracellular metabolism, potential modulatory roles, and unique synaptic specializations.

The synaptic polarity of bipolar cells is primarily determined by the type of glutamate receptor expressed on their dendrites. ON bipolar cells express metabotropic glutamate receptor mGluR6, which leads to depolarization in response to light—specifically when photoreceptor glutamate release diminishes. In contrast, OFF bipolar cells express ionotropic AMPA and kainate receptors, resulting in hyperpolarization under the same conditions, thereby encoding light decrements [4,20,29]. Table 1 provides a summary of the key neurochemical features of retinal bipolar cells.

At the axon terminals, located in the inner plexiform layer, bipolar cells express receptors for multiple neuromodulators, including GABA (A, B, and C subtypes), dopamine (D1 receptors), and glycine. These receptors mediate input from amacrine cells and contribute to the shaping of bipolar output through inhibitory and modulatory circuits [18].

Retinal bipolar cells demonstrate distinct amino acid profiles that correlate with their cellular localization and subtype identity. Glutamine is notably enriched in displaced bipolar cells, located within the ganglion cell layer, while aspartate is detected in both displaced and conventionally placed bipolar cells. Conversely, GABA and glycine are found only in a subset of bipolar cells in the inner nuclear layer and are absent from displaced bipolar cells, suggesting compartmentalized neurochemical roles based on cell positioning and connectivity [4,20,29]. These findings imply that, beyond their classical role as neurotransmitters, amino acids may also serve metabolic or neuromodulatory functions in bipolar cells, contributing to cell maintenance and synaptic regulation.

The neurochemical diversity among bipolar cells reflects cell-type-specific patterns of signaling and metabolism. The distribution of amino acids such as glutamine, aspartate, GABA, and glycine supports the existence of unique metabolic demands and synaptic specializations across bipolar subtypes [20]. These variations may influence the cells’ susceptibility to stress, injury, or degenerative processes. Moreover, bipolar cell connectivity with amacrine cells—via both chemical synapses and gap junctions—varies considerably across types, influencing their integration into retinal circuits. Differences in the number, type, and location of these synaptic contacts further underscore the functional diversity of bipolar cell populations, particularly in relation to their role in temporal processing, contrast sensitivity, and adaptation mechanisms [24,29].

## 5. Horizontal and Amacrine Cell Inputs in Bipolar Cell Processing

Lateral inhibition mediated by horizontal cells plays a critical role in spatial contrast enhancement within the retina. These interneurons form lateral connections with both photoreceptors and bipolar cells, enabling the suppression of adjacent photoreceptor signals. This process sharpens the spatial resolution of visual stimuli and underlies the center-surround organization of bipolar cell receptive fields [29].

In addition to their inhibitory output, horizontal cells engage in feedback mechanisms that modulate photoreceptor neurotransmission. Feedback to photoreceptors can occur through mechanisms such as hemi gap junctions or alterations in the extracellular ionic environment—particularly shifts in pH or calcium—which affect photoreceptor glutamate release and, by extension, influence bipolar cell activation [29]. This feedback ensures dynamic modulation of the bipolar cell response, particularly under varying luminance and spatial conditions. The interplay between direct photoreceptor input and lateral inhibition from horizontal cells is essential for the generation of the center-surround receptive fields characteristic of bipolar cells. These receptive fields enable the encoding of local contrast differences, a foundational step in spatial vision [20,29].

Table 2 provides a summary of signal integration in the inner and outer plexiform layers.

Amacrine cells, located in the inner retina, provide critical lateral and feedback modulation at the level of bipolar cell axon terminals and ganglion cells. Through their synaptic contacts, they modulate the flow of information between bipolar and ganglion cells, thereby refining visual signals before they exit the retina [28,30].

A hallmark of many amacrine cell synapses is their reciprocal arrangement with bipolar cell ribbon synapses. In these configurations, amacrine cells provide immediate feedback to the same bipolar terminal from which they receive input, enabling the localized and temporally precise modulation of synaptic transmission. Functionally, amacrine cells are highly diverse and employ neurotransmitters such as GABA and glycine to regulate temporal dynamics, contrast sensitivity, and signal integration. Their influence is crucial for features such as motion detection, transient response shaping, and luminance adaptation [28,30].

Within the rod pathway, specialized amacrine cells—including AII and A17 types—play indispensable roles in low-light (scotopic) vision. AII amacrine cells serve as key intermediaries in the transmission of signals from rod bipolar cells to cone bipolar pathways and ultimately to ganglion cells, while A17 cells modulate rod bipolar cell output through feedback inhibition [28]. These interactions ensure signal fidelity and gain control in dim lighting conditions.

## 6. Synaptic Gain in the Retina

Synaptic gain in the retina—the modulation of the signal strength during synaptic transmission—is a dynamic process essential for visual adaptation. At the bipolar cell synapse, presynaptic mechanisms such as synaptic vesicle depletion and voltage-gated calcium channel inactivation contribute to the activity-dependent regulation of neurotransmitter release. These mechanisms allow retinal circuits to adjust the signal amplitude in response to sustained changes in luminance and contrast, thereby preventing synaptic saturation and preserving the sensitivity to new stimuli [31,32].

Glutamate serves as the principal excitatory neurotransmitter in the retina, mediating signal transmission at both photoreceptor-to-bipolar and bipolar-to-ganglion cell synapses. As discussed above, bipolar cells are functionally divided into ON and OFF types, with ON bipolar cells expressing the metabotropic glutamate receptor mGluR6 to mediate depolarizing responses to decreased glutamate release, while OFF bipolar cells express ionotropic AMPA and kainate receptors that mediate hyperpolarizing responses. These receptor types not only determine the polarity of bipolar cell responses but also modulate synaptic gain, influencing how retinal circuits encode temporal and contrast features of the visual environment [32,33,34].

Recent evidence highlights that gain modulation is distributed across multiple synaptic layers of the retina. Depressive mechanisms at excitatory synapses, such as those between bipolar cells and ganglion cells, reduce gain and contribute to visual adaptation. Conversely, depression at inhibitory synapses from amacrine cells onto bipolar cells can result in disinhibition, effectively increasing the gain. These context-dependent adjustments in synaptic gain are critical in maintaining sensitivity to weak stimuli while filtering out redundant information, thus optimizing visual coding under variable environmental conditions [32,33,34].

## 7. Horizontal Cells

HCs participate in the slow surround inhibition of photoreceptors, BCs, and RGCs through noncanonical signaling pathways. Several models have been proposed, including ephaptic signaling via hemichannels, synaptic pH modulation, and transporter-mediated mechanisms, although the precise molecular basis remains unresolved. The sustained nature of HC feedback suggests that traditional vesicular release is insufficient to account for its efficacy [35,36].

These interneurons integrate and regulate input from multiple photoreceptors, contributing to contrast enhancement and spatial processing. They primarily release the inhibitory neurotransmitter GABA, which interacts with GABA receptors on photoreceptor terminals to modulate their activity [4].

Horizontal cells are primarily GABAergic interneurons in the majority of vertebrate retinas, characterized by their synthesis, storage, and release of γ-aminobutyric acid (GABA) [4,37,38]. This inhibitory neurotransmitter plays a key role in shaping retinal signal transmission. However, the presence of GABA in horizontal cells exhibits species-specific variability; for instance, its expression in primate horizontal cells remains a topic of ongoing debate [4]. GABA released by horizontal cells can exert autaptic effects, modulating their own membrane potential, and may also influence ionic dynamics within the synaptic cleft, thereby affecting photoreceptor signaling [39].

A core function of horizontal cells is to mediate inhibitory feedback to photoreceptors (rods and cones), a process that is integral to lateral inhibition and the formation of center-surround receptive fields in ganglion cells [38,40]. This feedback sharpens spatial contrast and enables the dynamic adaptation of the retinal output under varying luminance conditions, enhancing overall visual perception [40,41]. Although horizontal cells are GABAergic, GABA release may not be the dominant mechanism of cone inhibition. Instead, two alternative, noncanonical mechanisms have been proposed.
Ephaptic Inhibition: A rapid, nonsynaptic form of inhibition mediated through extracellular potential changes at the invaginating synapse between horizontal cells and cones. This ephaptic mechanism alters the cone membrane potential and modulates neurotransmitter release without traditional synaptic transmission [4].pH-Mediated Feedback: A slower, chemically mediated mechanism involving ATP release from horizontal cell dendrites via Pannexin 1 channels. This release leads to extracellular acidification, which suppresses voltage-gated Ca^2+^ channel activity in cones, thereby reducing glutamate release and modulating synaptic gain [4,39].

Horizontal cells display molecular and functional heterogeneity, with distinct subtypes identified across species. In primates, two morphologically and functionally distinct classes—H1 and H2 horizontal cells—exhibit differential connectivity with specific cone subtypes and spectral tuning, contributing to chromatic processing [40,41]. These subtypes are arranged in a nonrandom mosaic across the retinal surface, ensuring uniform coverage and functional integration [4,41]. By integrating input from multiple photoreceptors, horizontal cells facilitate both localized and long-range interactions in the outer retina. Their inhibitory influence is essential for contrast enhancement, spatial filtering, and the establishment of color opponency mechanisms. Furthermore, horizontal cells contribute to the adaptation of retinal circuits under changing illumination, ensuring optimal encoding of the visual environment [4,38,40,41].

## 8. Amacrine Cells

ACs and axonal cells (AxCs) are the only retinal neurons that establish synaptic contacts resembling conventional CNS gray matter synapses. ACs predominantly form inhibitory synapses utilizing GABA and glycine, although some subpopulations also co-release acetylcholine and neuropeptides. The primary targets of AxCs remain incompletely characterized but likely include ACs and retinal ganglion cells (RGCs). Unlike ribbon synapses, AC synapses are monadic and exhibit precise presynaptic-to-postsynaptic targeting akin to multipolar neurons in the CNS [2,42].

Amacrine cells are diverse interneurons that modulate synaptic transmission between bipolar cells and ganglion cells. They utilize various neurotransmitters, summarized in Table 3.

Amacrine cells represent one of the most neurochemically and functionally diverse neuronal populations in the vertebrate retina. These interneurons play essential roles in the integration, modulation, and refinement of visual signals, operating at the level of the inner plexiform layer to influence bipolar-to-ganglion cell transmission. Their heterogeneity, both molecular and functional, is fundamental to shaping complex visual computations including contrast sensitivity, motion detection, and adaptation to changing light conditions.

The vast majority of amacrine cells are inhibitory, utilizing γ-aminobutyric acid (GABA) or glycine as their primary neurotransmitters [4,43]. These inhibitory signals sculpt the spatial and temporal dynamics of retinal output through synapses with bipolar and ganglion cells, regulating gain control and the receptive field properties [4,44]. In addition to classical inhibitory transmitters, amacrine cells exhibit remarkable neurochemical diversity.

Dopaminergic amacrine cells release dopamine, serving as neuromodulators that influence retinal adaptation to ambient illumination and mediate the circadian modulation of retinal circuits [4,43]. Starburst amacrine cells, which are cholinergic, release acetylcholine and are central to the computation of direction-selective responses, particularly in motion-sensitive pathways [43,45]. Moreover, subsets of amacrine cells release nitric oxide and other unconventional neuromodulators, expanding the repertoire of chemical signals used for local circuit modulation.

Recent advances in transcriptomics and functional imaging have uncovered an extraordinary range of amacrine cell types. A 2025 comprehensive study identified over 40 physiologically distinct GABAergic amacrine cell types in the mouse retina, each characterized by unique synaptic release kinetics and tuning for specific visual features. Molecular clustering has revealed at least 63 discrete amacrine cell classes, with GABAergic cells constituting approximately 70% of this population [4]. Table 4 summarizes the amacrine cells, which display a wide range of dendritic morphologies, from narrow-field cells involved in local signal modulation to wide-field cells that integrate information over larger retinal areas. These cells are organized in a nonrandom mosaic distribution, ensuring the uniform sampling of the visual space across the retina. Functionally, amacrine cells form the bulk of the synaptic interactions within the inner plexiform layer, where they modulate the bipolar cell output before transmission to retinal ganglion cells [43,44,46].

Their integration into diverse retinal microcircuits enables the retina to perform the complex preprocessing of visual information prior to central transmission, underscoring the essential role of amacrine cells in visual computation.

## 9. Retinal Ganglion Cells (RGCs)

RGCs are the output neurons of the retina, transmitting visual information to the brain. They receive excitatory input via glutamate acting on ionotropic receptors (AMPA, kainate, and NMDA receptors). Additionally, intrinsically photosensitive retinal ganglion cells (ipRGCs) contain melanopsin and can respond directly to light. These ipRGCs also receive synaptic inputs from bipolar and amacrine cells, integrating both light-dependent and synaptic signals. Understanding these synaptic interactions and neurotransmitter dynamics is essential in elucidating retinal function and developing treatments for visual disorders.

Retinal ganglion cells (RGCs) serve as the principal output neurons of the retina, encoding processed visual signals and transmitting them to the brain via the optic nerve. Their activity reflects a finely tuned balance of excitatory and inhibitory synaptic inputs, mediated by a diverse array of neurotransmitters. This neurochemical complexity underlies the heterogeneous response properties and parallel visual pathways conveyed by different RGC subtypes [48].

The dominant excitatory drive to RGCs is glutamatergic, originating from bipolar cells at sign-conserving synapses in the inner plexiform layer [49]. Depending on the polarity of the bipolar cell input—ON or OFF—ganglion cells exhibit corresponding light-evoked depolarization or hyperpolarization, forming the basis of center-surround receptive field organization [49]. In addition to excitatory input, RGCs receive robust inhibitory modulation from amacrine cells via the neurotransmitters γ-aminobutyric acid (GABA) and glycine [28,50]. These inhibitory circuits are essential in shaping the temporal precision, contrast sensitivity, and spatial tuning of ganglion cell responses. Beyond classical excitatory and inhibitory transmitters, select RGC populations are influenced by acetylcholine, notably in direction-selective circuits. In these pathways, cholinergic excitation from starburst amacrine cells plays a critical role in encoding motion direction, particularly in subsets of ON–OFF direction-selective ganglion cells [4,49].

A small subset of RGCs is intrinsically photosensitive, expressing the photopigment melanopsin. These intrinsically photosensitive retinal ganglion cells (ipRGCs) are capable of directly responding to light and contribute to non-image-forming visual functions, including circadian photoentrainment and pupillary light reflexes. RGCs encompass multiple anatomically and functionally distinct classes, such as midget (P-type), parasol (M-type), and small bistratified (K-type) cells, each characterized by unique receptive field structures, synaptic inputs, and projection targets within central visual pathways [49]. Their diversity is further amplified by the subtype-specific integration of vertical excitatory input from photoreceptor-to-bipolar pathways and lateral modulation from horizontal and amacrine cells [48]. This convergence of synaptic signals enables RGCs to perform complex computations, including edge detection, motion discrimination, and contrast encoding, prior to relaying information to higher-order visual centers.

## 10. Slow Neurotransmitter Signaling in the Retina

In addition to classical synaptic transmission, retinal neurons engage in volume transmission mediated by dopamine, serotonin, histamine, and neuropeptides. These neurotransmitters are released in a nonfocal manner and modulate retinal circuits through G-protein-coupled receptors (GPCRs). In nonmammalian species, efferent inputs from the CNS provide fast synaptic input to ACs, often through GABAergic signaling, whereas mammalian retinal efferents primarily employ volume transmission. Recent studies highlight the importance of slow neurotransmitter signaling in the temporal shaping and adaptive tuning of retinal circuits. These mechanisms, mediated by asynchronous GABA release, slow-acting neuromodulators, and volume transmission, operate across both developing and mature retinas to regulate circuit excitability, signal timing, and network coordination [51,52,53,54,55].

A subset of amacrine cells (ACs) release GABA asynchronously onto rod bipolar cells (RBCs), generating prolonged inhibitory postsynaptic currents (IPSCs). This slow GABAergic transmission is triggered by sustained calcium elevation within presynaptic terminals, primarily through L-type Ca^2+^ channels and intracellular Ca^2+^ stores [51,53]. This form of inhibition complements the slow, graded glutamate release from bipolar cell ribbon synapses, ensuring temporal alignment between excitatory and inhibitory inputs in scotopic pathways.

The kinetics of inhibition are further tuned by receptor subtype engagement: GABA_C receptor-mediated currents are the slowest, while GABA_A and glycine receptor-mediated responses are comparatively rapid. The differential distribution of these receptor subtypes across retinal pathways enables the precise modulation of inhibitory timing to match the functional demands of specific circuits [51,53].

In the developing retina, slow neurotransmission contributes to the regulation of neurogenesis, gliogenesis, and circuit formation. Neuromodulators such as ATP and acetylcholine, acting through metabotropic and purinergic receptors (e.g., muscarinic and P2 receptors), elicit long-duration calcium transients in Müller glia and immature neurons. These signals promote neuron–glia communication and synchronize activity across wide retinal domains, supporting the coordinated maturation of retinal networks [52,56].

Slow neuromodulatory signaling also underlies adaptive changes in retinal output. The gradual hyperpolarization of retinal ganglion cells (RGCs), for example, has been linked to sustained modulatory inputs that adjust the membrane potential and synaptic efficacy over seconds to minutes. Such slow adaptation mechanisms enable the retina to maintain sensitivity across varying lighting conditions and visual contexts [54,55].

In addition to classical synaptic release, slow neurotransmission in the retina can occur via volume transmission. In this mode, diffusible transmitters such as dopamine and ATP are released into the extracellular space and act on distant targets, modulating the circuit excitability and synaptic strength without the need for direct synaptic contact. These nonsynaptic interactions introduce a slower, spatially diffuse layer of modulation that is critical for global retinal functions, including light adaptation and circadian regulation [52,55].

## 11. Neuromodulatory Systems: Dopamine and Acetylcholine

Dopaminergic amacrine cells play a crucial role in light adaptation, modulating gap junction coupling and synaptic gain. Dopamine acts through D1 and D2 receptors to adjust the sensitivity of bipolar and horizontal cells, reducing coupling under bright conditions to enhance the spatial resolution. Acetylcholine, released by starburst amacrine cells, contributes to directional selectivity and motion processing in the inner retina [57,58].

Extensive evidence has established that neuromodulators such as dopamine and acetylcholine play essential and highly specialized roles in the retina. These modulators fine-tune the retinal circuitry by influencing synaptic transmission, cellular excitability, and adaptive responses to dynamic visual environments. Recent studies continue to uncover the cell-type-specific and context-dependent actions of neuromodulators, underscoring their importance in shaping the retinal output [59,60]. Table 5 summarizes the retinal neuromodulators and their source cells.

Dopamine, primarily released by a subset of amacrine cells, is a central modulator of retinal function under photopic (daylight) conditions. Light stimulation triggers dopamine release, which acts on D1-like and D2-like dopamine receptors expressed on multiple retinal cell types, including photoreceptors, bipolar cells, horizontal cells, and ganglion cells [59]. Through its widespread receptor distribution, dopamine orchestrates a shift from rod-dominated scotopic processing to cone-dominated photopic signaling. It achieves this by modulating electrical coupling via gap junctions, particularly among horizontal and amacrine cells; altering the sensitivity and receptive field properties of bipolar and ganglion cells; and regulating neurotransmitter release and reuptake, particularly GABA and glutamate [59]. Dopamine also plays a role in circadian regulation, aligning retinal function with daily light–dark cycles.

Acetylcholine is predominantly released by starburst amacrine cells, a well-characterized interneuron class that is critical for motion detection. Cholinergic signaling in the retina involves both nicotinic and muscarinic receptors, which are expressed on retinal ganglion cells and other amacrine cells [59]. The key roles of acetylcholine in retinal circuits include establishing direction selectivity, particularly in direction-selective ganglion cells; modulating the excitability and spike timing of ganglion cells in response to moving stimuli; and contributing to the activity-dependent development of visual circuits during early postnatal stages [59].

Beyond classical neurotransmitters, several additional neuromodulatory systems operate in the retina. Nitric oxide (NO), for instance, acts as a diffusible signaling molecule that modulates synaptic function and retinal excitability. Recent findings demonstrate that NO selectively modulates contrast suppression and temporal tuning in specific ganglion cell types without affecting their spatial tuning properties [60].

Retinoic acid, another emerging retinal modulator, has been implicated in both circuit development and plasticity. Together, these modulators expand the retina’s capacity for nonsynaptic communication and the long-range coordination of neural activity [59,60]. A comprehensive 2023 review compared dopamine and retinoic acid as modulators of retinal activity, highlighting their overlapping and distinct roles in synaptic modulation and adaptive signaling. The review also emphasized the increasing recognition of nitric oxide and other diffusible modulators as integral to the regulation of retinal output and visual sensitivity under changing environmental conditions [59].

## 12. Gap Junctional Coupling in the Retina

Intercellular coupling is a pervasive feature of the retinal circuitry, occurring both homocellularly (within the same cell type) and heterocellularly (between different cell types) via gap junctions. Dopaminergic modulation plays a critical role in regulating gap junction conductance, with D1 receptor activation decreasing HC–HC and AC–AC coupling, while D2 receptor activation modulates rod–cone coupling. Recent ultrastructural analyses indicate that heterocellular coupling is common between ACs and RGCs, with specific coupling patterns observed between ON and OFF subtypes [61,62].

Extensive research has established gap junctions as fundamental components of the retinal circuitry, mediating electrical coupling and intercellular signaling across multiple neuronal classes. These specialized intercellular channels contribute to visual processing by facilitating synchrony, spatial integration, and adaptive plasticity in both healthy and diseased retinal states. Gap junctions form electrical synapses that enable the direct transfer of ions and small signaling molecules between adjacent neurons. Unlike chemical synapses, they support rapid and bidirectional communication, which is critical for the coordination of activity across retinal networks [63,64]. Gap junctions are broadly expressed across all major retinal cell types—including photoreceptors (rods and cones), horizontal cells, bipolar cells, amacrine cells, and ganglion cells—where they participate in various computational functions, such as the synchronization of neuronal firing, signal averaging and spatial pooling, and the enhancement of the signal to noise ratio [63,65].

Connexin 36 (Cx36) is the principal neuronal connexin expressed in the retina and is particularly enriched at synapses between rods and cones, as well as within AII amacrine cell networks involved in scotopic (low-light) vision [63,65]. Rod–cone coupling via Cx36-mediated gap junctions allows rod-driven signals to access cone pathways, thereby extending visual sensitivity under dim light conditions. Each cone typically forms dozens of gap junctions with neighboring rods, creating a robust substrate for mixed photoreceptor signaling [65]. In addition to neuronal circuits, gap junctions are present in the retinal pigment epithelium (RPE), where they contribute to epithelial integrity, ion homeostasis, and intercellular signaling [64].

Retinal gap junctions exhibit dynamic regulation, modulated by factors such as the ambient illumination, circadian phase, and neuromodulatory input—particularly dopamine. Dopaminergic signaling reduces the Cx36 coupling strength via D1-like receptor activation, thereby decoupling photoreceptors and amacrine cells to refine the receptive field properties under photopic conditions [63,66]. This modulation enables flexible circuit reconfiguration across multiple timescales. The rapid (milliseconds to seconds) process effects light-driven changes in the receptive field size, while the slow (hours to days) process has its effect during circadian transitions in network connectivity [63,66].

During retinal development, gap junctions mediate spontaneous activity waves across retinal ganglion cells, contributing to the refinement of visual circuits and eye-specific segregation in central targets [62]. In mature retinas, these channels remain essential for coordinated activity and spatial integration. Disruptions in gap junction coupling—due to genetic mutations, metabolic stress, or inflammation—have been linked to a range of retinal pathologies, including degenerative diseases affecting photoreceptors and the RPE [64]. Understanding the role of electrical synapses in these conditions may offer new therapeutic targets for the preservation or restoration of retinal function.

## 13. Conclusions

The diversity of the synaptic architectures and neurotransmission mechanisms in the retina underscores its complexity as a neural processing system. Ribbon synapses enable high-speed vesicle release and sustain neurotransmission, while conventional and volume transmission mechanisms further refine retinal signaling. Intercellular coupling and glutamate receptor diversity contribute to the dynamic modulation of retinal circuits, ensuring the precise encoding of visual information. Continued research into these processes will enhance our understanding of synaptic function in both health and disease.

## Figures and Tables

**Figure 1 brainsci-15-00727-f001:**
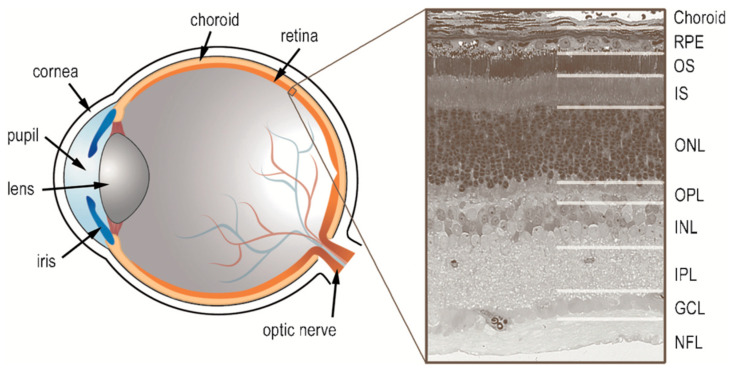
Schematic of the eye and a corresponding micrograph of the retina with retinal layers indicated. The eye structures are labeled and the retinal layers indicated as follows: retinal pigment epithelium (*RPE*); outer segments of photoreceptors (*OS*); inner segments of photoreceptors (*IS*); outer nuclear layer (*ONL*), which is composed primarily of photoreceptor cell nuclei; outer plexiform layer (*OPL*), which is the site for synaptic contact between photoreceptors, horizontal cells, and bipolar cells; inner nuclear layer (*INL*), the location of cell bodies of bipolar cells and most horizontal and amacrine cells; inner plexiform layer (*IPL*), the site of synaptic contact between bipolar cells, amacrine cells, and ganglion cells; ganglion cell layer (*GCL*), the location of most ganglion cell somata; and nerve fiber layer (*NFL*), comprised of the axons of ganglion cells.

**Figure 2 brainsci-15-00727-f002:**
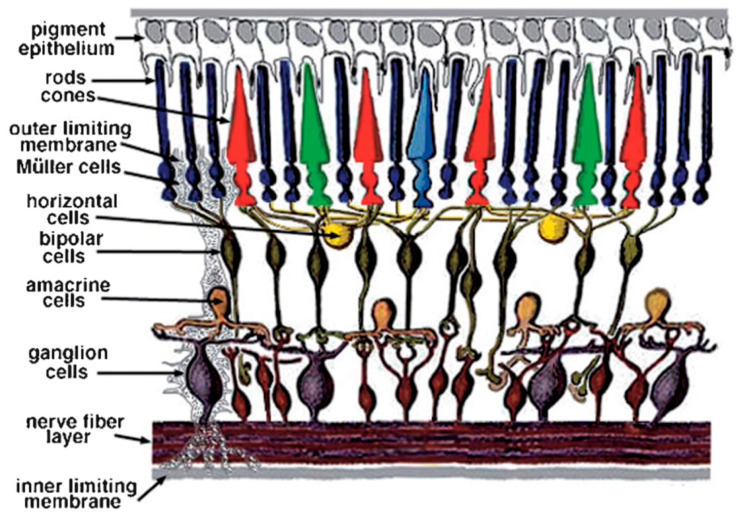
Organization of the vertebrate retina, showing the cell types in their respective layers. Schematic courtesy of Dr. Helga Kolb.

**Table 1 brainsci-15-00727-t001:** Retinal bipolar cell neurochemical differences.

Feature	ON Bipolar Cells	OFF Bipolar Cells
Receptor Activation	Darkness	Light
Main Neurotransmitter	Glutamate	Glutamate
Dendritic Receptor Types	mGluR6, metabotropic	AMPA/kainate (ionotropic)
Axonal Receptor Types	GABA (A, B, C), glycine, D1	GABA (A, B, C), glycine, D1
Amino Acid Content	May contain aspartate	May contain aspartate
Synaptic Integration	Chemical and electrical	Chemical and electrical

**Table 2 brainsci-15-00727-t002:** OPL/IPL signal integration.

Source	Layer	Mechanism/Effect
Horizontal	Outer plexiform	Lateral inhibition, feedback, center-surround
Amacrine	Inner plexiform	Lateral inhibition, temporal/contrast modulation

OPL = outer plexiform layer; IPL = inner plexiform layer.

**Table 3 brainsci-15-00727-t003:** Amacrine cell neurotransmitters.

Neurotransmitter	Function
GABA	Inhibitory, affects signal integration and temporal aspects of vision
Glycine	Inhibitory
Dopamine	Modulates retinal processing under varying light conditions

**Table 4 brainsci-15-00727-t004:** Functions of amacrine cells.

Amacrine Cell Function	Source
Inhibitory and reciprocal synaptic modulation of bipolar and ganglion cell activity	[46,47]
Creation of receptive field surrounds and implementation of temporal filtering mechanisms	[47]
Direction-selective signal processing via starburst amacrine cells	[43]
Signal transmission in the rod pathway under scotopic conditions, primarily mediated by AII amacrine cells	[4]
Light adaptation and circadian regulation, orchestrated by dopaminergic amacrine cells	[4,43]

**Table 5 brainsci-15-00727-t005:** Neuromodulators in the retina.

Neuromodulator	Source Cells	Principal Effects
Dopamine	Dopaminergic amacrines	Light adaptation, circadian modulation, gap junction uncoupling
Acetylcholine	Starburst amacrines	Direction selectivity, motion detection, RGC excitation
Nitric oxide	Amacrines, others	Modulation of contrast, timing of RGC responses
Retinoic acid	Various	Developmental signaling, synaptic modulation

RGC = retinal ganglion cell.

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
