# Peer review of "Retinal Neurochemistry"

_brainsci, 2025, doi:10.3390/brainsci15070727_

Round 1

Reviewer 1 Report

Comments and Suggestions for Authors

This review provides a broad overview of retinal biology with discussion of relevant mediators involved in vision. This topic has been well-reviewed in the literature. Clarification in the introduction regarding the purpose of this review over previously published papers on this topic should be added. Also, this review would benefit from a Figure representing the mechanisms underlying retinal signaling, as well as significant editing of the Tables to improve their usefulness. Specific comments are listed below.

-In general, the tables are very minimalistic and not terribly useful to the reader in providing critical comparisons between groups (Table 2 - OPL vs IPL and Table 3 - Neurotransmitters)

-Based on Table 1, the only difference between ON and OFF BP cells is the dendritic receptor, which is not completely accurate given the known signaling pathways associated with light vs dark signaling. OFF BP cells are activated by darkness; ; there are also differences in regards to function. Further clarification should be added to the tables with a comparison of differences. Moreover, there is little mention in the text of the light intensity differences in ON vs OFF BP in terms of contrast.

-It is assumed that the authors are referring to the mammalian retina in their descriptions, but it is not completely clear given that some references are in other organisms (ex: lizard?). There are also differences in humans versus other mammals. This should be clarified in the Introduction.

-Table (5) is mis-numbered as Table 1.

-The abbreviation list at the end is definitely not comprehensive and should be updated to include all abbreviations mentioned (ex: CX3C, iPSCs)

Author Response

Review 1

This review provides a broad overview of retinal biology with discussion of relevant mediators involved in vision. This topic has been well-reviewed in the literature. Clarification in the introduction regarding the purpose of this review over previously published papers on this topic should be added. Also, this review would benefit from a Figure representing the mechanisms underlying retinal signaling, as well as significant editing of the Tables to improve their usefulness. Specific comments are listed below.

We have added 2 figures to provide visual clarity as to the terminology. Figure 1 shows the retinal layers indicated on a photomicrograph. Figure 2 is a retinal schematic to show the organization of the major cell types. We believe this may help address the mechanistic questions and tie together the tables.

-In general, the tables are very minimalistic and not terribly useful to the reader in providing critical comparisons between groups (Table 2 - OPL vs IPL and Table 3 - Neurotransmitters)

The tables are intended as short summaries of the sections. Tables 2 and 3 in particular are minimal to illustrate the key points. Table 4 is to provide reference for the amacrine functions, with Table 5 summarizing the neuromodulators. We have retained them, and modified Table 1 as below.

-Based on Table 1, the only difference between ON and OFF BP cells is the dendritic receptor, which is not completely accurate given the known signaling pathways associated with light vs dark signaling. OFF BP cells are activated by darkness; ; there are also differences in regards to function. Further clarification should be added to the tables with a comparison of differences. Moreover, there is little mention in the text of the light intensity differences in ON vs OFF BP in terms of contrast.

We modified Table 1 to address this point, and thank you for suggesting it, as it makes the table a better summary of the bipolar cell section.

-It is assumed that the authors are referring to the mammalian retina in their descriptions, but it is not completely clear given that some references are in other organisms (ex: lizard?). There are also differences in humans versus other mammals. This should be clarified in the Introduction.

The introduction now makes clear that the work is about vertebrate retina, and draws from all studies to elucidate mammalian retinal function.

-Table (5) is mis-numbered as Table 1.

Thank you. This has been corrected now.

-The abbreviation list at the end is definitely not comprehensive and should be updated to include all abbreviations mentioned (ex: CX3C, iPSCs)

We have updated the abbreviation list, and re-checked to ensure all terms are now included.

Reviewer 2 Report

Comments and Suggestions for Authors

Dear editor and authors,

This is a comprehensive and well-structured review of the neurochemical architecture of the retina, with a focus on glutamatergic and GABAergic systems across distinct retinal cell types. The manuscript is clearly relevant for Brain Sciences and contributes an updated synthesis of a highly complex area of visual neuroscience so i reccomend to accept it after minor revision.

The English language is appropriate and understandable.

The manuscript is presented in a well structured manner.

References do not comprise self citations. The references are not within last 5 years but that is due to the specific rarity of the case presented.

Table 1 is used twice (lines 173 and 451). The second instance should be relabeled as Table 5 to maintain numbering consistency.

A schematic diagram of the retina (even if schematic or adapted) would enhance comprehension, especially for readers outside the retinal field.

Author Response

Review 2

Dear editor and authors,

This is a comprehensive and well-structured review of the neurochemical architecture of the retina, with a focus on glutamatergic and GABAergic systems across distinct retinal cell types. The manuscript is clearly relevant for Brain Sciences and contributes an updated synthesis of a highly complex area of visual neuroscience so i reccomend to accept it after minor revision.

The English language is appropriate and understandable.

The manuscript is presented in a well structured manner.

References do not comprise self citations. The references are not within last 5 years but that is due to the specific rarity of the case presented.

Table 1 is used twice (lines 173 and 451). The second instance should be relabeled as Table 5 to maintain numbering consistency.

This has been rectified

A schematic diagram of the retina (even if schematic or adapted) would enhance comprehension, especially for readers outside the retinal field.

A schematic of the retina is now included as Figure 2, and we have also added a labelled retinal image to denote where the specific layers are located. Thank you for this recommendation as we believe it helps the readability of the review.

Reviewer 3 Report

Comments and Suggestions for Authors

The manuscript by Dominic Man-Kit Lam and George Ayoub is a comprehensive up-to-date review on retinal neurochemistry with the emphasis on the diversity of neurotransmitter and neuromodulatory systems in different retinal cell types. Such reviews are not frequent and this one will be interesting to general readership. I have only a couple of comments and suggestions:

  1. The manuscript has no figures, only tables. A scheme of the retinal cell types with the major neurotransmitters and receptors involved in the signal processing will be very helpful to visualize the data for the reader.
  2. There are some redundant paragraphs in the text. For instance, the lines 37-40 mostly repeat the information from the previous paragraph. The lines 246-251 also repeat the earlier information, mostly from the lines 166-171. Please check and correct if necessary.
  3. With the major role of glutamate and its different receptor types in the retina it might be useful to include the information that AMPA receptors are traditionally divided into two  subtypes: calcium-permeable (CP-AMPARs) and calcium-impermeable (CI-AMPARs). And that CP-AMPARs have their unique roles in the retina. Please see Diamond, 2011, Front Mol Neurosci for review. 

Author Response

Review 3

The manuscript by Dominic Man-Kit Lam and George Ayoub is a comprehensive up-to-date review on retinal neurochemistry with the emphasis on the diversity of neurotransmitter and neuromodulatory systems in different retinal cell types. Such reviews are not frequent and this one will be interesting to general readership. I have only a couple of comments and suggestions:

  1. The manuscript has no figures, only tables. A scheme of the retinal cell types with the major neurotransmitters and receptors involved in the signal processing will be very helpful to visualize the data for the reader.

New figures have been added to address this point. Figures 1 and 2 provide a image of a retinal slice with the layers indicated, and a schematic of the retina. We thank you for this recommendation and feel it may help visualize the review for the readers.

  1. There are some redundant paragraphs in the text. For instance, the lines 37-40 mostly repeat the information from the previous paragraph. The lines 246-251 also repeat the earlier information, mostly from the lines 166-171. Please check and correct if necessary.

We have reduced the redundancy in each of these, as indicated in the marked-up version of the manuscript.

  1. With the major role of glutamate and its different receptor types in the retina it might be useful to include the information that AMPA receptors are traditionally divided into two  subtypes: calcium-permeable (CP-AMPARs) and calcium-impermeable (CI-AMPARs). And that CP-AMPARs have their unique roles in the retina. Please see Diamond, 2011, Front Mol Neurosci for review. 

Thank you for this recommendation. We have added this important point in section 3, Bipolar cell receptors, in the first paragraph (lines 163-170)